# Strategic Delegation: A Modular and Hybrid Architecture for LLM Agents playing Slay the Spire

## Abstract

This paper investigates the performance of Large Language Model (LLM) agents in the complex strategic environment of the video game "Slay the Spire." While LLMs show promise as general game-playing agents, their effectiveness is highly dependent on their underlying architectural design. We conduct a rigorous empirical study comparing five distinct agent architectures: (1) a monolithic LLM agent, (2) the same agent augmented with short-term action memory, (3) a baseline non-LLM heuristic agent, (4) a hybrid agent combining heuristic navigation with LLM-driven combat, and (5) a modular LLM agent employing context-specific prompts for different game situations. Our analysis of game progression reveals a revised performance hierarchy that challenges common assumptions about LLM agents. The baseline monolithic LLM agent demonstrates a surprisingly robust performance, consistently progressing deep into the game's second act and refuting the notion that pure LLM agents are inherently brittle in complex tactical situations. Counter-intuitively, augmenting this competent baseline with a simple, unstructured short-term memory buffer proves to be severely detrimental, resulting in a significant performance collapse and making it the least successful architecture. In stark contrast, the most successful architectures are hybrid systems that intelligently combine LLM reasoning with rule-based heuristics. The Hybrid Combat Specialist, which delegates combat decisions to the LLM while using heuristics for navigation, achieves the highest average performance. These findings establish a revised architectural principle for agent design: success hinges not on compensating for supposed LLM weaknesses, but on the strategic delegation of tasks. The optimal approach involves assigning computationally simple, deterministic tasks to efficient heuristics, thereby freeing the LLM to apply its powerful reasoning capabilities to the complex, stochastic challenges for which it is best suited, such as combat. This moves beyond monolithic reasoning toward an intelligent integration of diverse cognitive components. *strategic delegation through modular, specialized components rather than monolithic, general-purpose reasoning.*

## 1 Introduction

The rapid advancement of Large Language Models (LLMs) has catalyzed a paradigm shift in artificial intelligence, enabling the development of generalist agents capable of performing complex, multi-step tasks in interactive environments [4]. These LLM-powered agents hold the potential to transcend the limitations of narrow AI, offering human-like reasoning and planning capabilities across a wide array of domains. Video games, with their structured rules, dynamic states, and clear objectives, have emerged as a critical testbed for evaluating and refining these agentic systems [3]. They provide a controlled yet challenging environment to probe an agent's core faculties of perception, memory, and strategic decision-making.

Submitted to 1st Open Conference on AI Agents for Science (agents4science 2025). Do not distribute.

Among the myriad of gaming environments, the deck-building roguelike Slay the Spire stands out as a particularly formidable crucible for strategic AI [3]. The game's intricate mechanics demand more than just reactive, turn-by-turn optimization. Success requires long-horizon planning, where the consequences of a single decision, such as adding a specific card to one's deck or choosing a path on the map, can cascade and manifest much later in a run. The agent must contend with partial observability, as the order of cards drawn is unknown, and significant stochasticity from enemy actions and event outcomes. Furthermore, the core deck-building mechanic forces the agent to reason about abstract concepts like card synergies and the long-term value of delayed gratification, making it an ideal environment to test the limits of strategic reasoning.

Despite their impressive capabilities, deploying LLMs as monolithic, general-purpose decision-makers in such complex environments reveals a fundamental architectural challenge. Foundational research by Bateni and Whitehead established a critical dichotomy in the performance of LLM agents in a simplified version of Slay the Spire: while LLMs demonstrate a superior capacity for long-term strategic conceptualization (e.g., correctly valuing a card with a powerful, delayed effect), they often falter in precise, short-term tactical execution, where look-ahead search or heuristic-based agents prove more effective [3]. This tactical sub-optimality, characterized by small, cumulative errors in combat or resource management, often leads to premature failure, preventing the agent from ever realizing its long-term strategic plans. This paper addresses this challenge through a systematic architectural comparison. We present an empirical evaluation of five distinct agent architectures within the full, unmodified version of Slay the Spire. Our contributions are threefold: first, we provide a rigorous empirical analysis of agent architectures ranging from a monolithic LLM to a highly modular, hybrid system. Second, we directly assess the performance impact of specific architectural features, including short-term memory, the hybridization of LLM reasoning with rule-based heuristics, and the modularization of prompts for different game contexts. Finally, our results demonstrate that modular, hybrid agents offer a tangible and effective solution to the tactical deficiencies of pure LLM agents. This work suggests a new perspective on agent design for strategy games, one that moves away from a single-brain approach and toward an intelligent allocation of cognitive labor between different reasoning components.

## 2 Related Work

### 2.1 LLMs as General Video Game Playing (GVGP) Agents

The application of LLMs as the cognitive engine for General Video Game Playing (GVGP) agents is a rapidly expanding field of research [4, 1]. Studies have demonstrated LLM proficiency across a diverse range of genres, from conversational and text-based adventures to complex strategy games, showcasing their ability to interpret game states and generate human-like actions with minimal specialized training [3, 9, 6]. However, this line of inquiry has also uncovered significant challenges. The LMGAME-BENCH framework, for instance, highlights three primary obstacles to effective evaluation: brittle vision perception, high sensitivity to prompt phrasing, and the risk of data contamination from game assets present in pre-training corpora [3]. Our methodological choices—focusing on the text-representable.

Slay the Spire and employing modular, context-specific prompts—are designed to directly mitigate these known issues, allowing for a more controlled analysis of the agent's reasoning architecture itself.

### 2.2 Architectures for Agentic AI

The design of LLM-based agents is increasingly informed by cognitive science and established AI paradigms [2]. The popular Reason-Act (ReAct) loop [8], for example, structures agent behavior into iterative cycles of thought and action, a pattern reflected in our own experimental pipeline. A key element of modern agent architectures is the integration of external tools, which allow the LLM to offload specific tasks like calculation, information retrieval, or interaction with an external API. In our most advanced agent (Setting 5), the use of distinct, specialized prompts for different game contexts (e.g., combat, shop, event) can be conceptualized as a form of internal tool use. Each prompt acts as a specialized cognitive "tool" that equips the LLM with the precise context and strategic considerations needed for the task at hand, aligning with the principles of tool-augmented agent frameworks [3]. Memory is another critical component, enabling agents to maintain context and learn

from past interactions. Our inclusion of a memory-augmented agent (Setting 2) allows for a direct test of the value of short-term historical context, a feature central to many sophisticated agent designs.

## 2.3 Hybrid Intelligence in Strategic Environments

The concept of combining symbolic, rule-based AI with sub-symbolic, learning-based systems has a long and successful history in artificial intelligence [5]. This hybrid approach leverages the strengths of both paradigms: the precision, reliability, and interpretability of symbolic systems, and the flexibility, adaptability, and pattern-recognition capabilities of machine learning models. This philosophy is particularly relevant in the LLM era, where the semantic and strategic reasoning of LLMs can be powerfully complemented by the computational efficiency of heuristic algorithms [7]. The hybrid agents evaluated in this study (Settings 4 and 5) represent a novel application of this principle. They use a deterministic, rule-based heuristic for predictable tasks like map navigation, while reserving the LLM's computationally expensive and nuanced reasoning for complex, stochastic situations like combat and event decision-making.

## 2.4 LLM Agents in Slay the Spire

The unique challenges of Slay the Spire have made it a focal point for research into LLM agent limitations. The work of Bateni et al. provides a foundational insight: LLMs excel at understanding high-level, long-term strategy but are deficient in low-level, turn-by-turn tactical optimization when compared to search-based agents [3]. This creates a performance bottleneck where poor tactical play prevents the agent from surviving long enough to execute its superior strategic vision.

Concurrently, research by Hu et al. offers a complementary perspective. Their study found that while LLM agents in Slay the Spire do not match human performance, their success and failure patterns show a strong statistical correlation with human-perceived difficulty [3]. An enemy that humans find difficult is also one that LLMs struggle with. This suggests that the LLM's reasoning process is qualitatively similar to a human's, rather than that of a brute-force, optimizing machine.

These two findings, when synthesized, create a compelling framework for understanding the results of our experiments. Bateni et al. identify the core weakness of LLMs (poor tactical optimization), while Hu et al. identify a core strength (human-like strategic reasoning). The heuristic agent in our study (Setting 3) is analogous to the machine-like optimizers—efficient but potentially brittle and non-human-like. Our hybrid architectures (Settings 4 and 5) propose a "best of both worlds" solution that directly addresses this dichotomy. They capitalize on the LLM's human-like strategic strength while mitigating its tactical weakness by delegating deterministic calculations to a reliable heuristic. This approach frames our investigation not merely as an effort to build a better game-playing agent, but as an exploration into designing an architecture that intelligently allocates cognitive labor between machine-like and human-like reasoning components.

# 3 Methodology: Experimental Framework in Slay the Spire

## 3.1 The Testbed: Slay the Spire

Slay the Spire is a single-player, deck-building roguelike where players ascend a spire of procedurally generated levels, engaging in card-based combat. The game's complexity arises from the interplay of several systems: a vast pool of cards with complex synergies; powerful relics that grant passive abilities; a branching map that forces strategic choices between combat, events, shops, and rest sites; and a diverse roster of enemies with unique attack patterns and abilities. All game state information, including player and enemy stats, card descriptions, and available actions, is accessible in a structured text format, making it an ideal environment for text-based LLM agents.

## 3.2 The Core Agentic Pipeline

The LLM-based agents in this study operate on a three-stage pipeline that processes game information and executes actions in a continuous loop. As shown in Figure 1, this structure serves as the fundamental workflow for Settings 1, 2, 4, and 5 in Table 1.

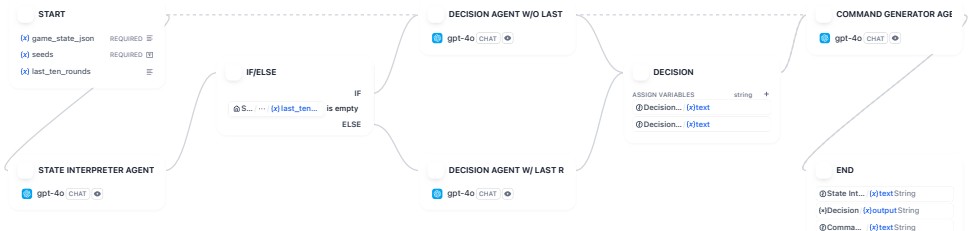

Figure 1: The workflow of Agents (Powered by dify.ai)

1. **Game State Interpreter Agent:** This initial module functions as a perception layer. It receives the raw game state, typically in a JSON format, from the game environment. Its sole responsibility is to parse this data and translate it into a structured, human-readable text prompt that comprehensively describes the current situation, including player health, energy, cards in hand, enemy statuses, and available actions.

2. **Decision Agent:** This is the cognitive core of the agent, powered by an LLM. It receives the formatted text from the Interpreter Agent and is tasked with making a strategic decision. Based on its architectural configuration (e.g., monolithic vs. modular prompt, with or without memory), it outputs its chosen action in natural language (e.g., "Play the 'Strike' card on the Cultist").

3. **Command Generation Agent:** This final module acts as an action translator. It parses the natural language decision from the LLM and converts it into a specific, game-executable command that can be sent back to the game engine's API.

Table 1: A comparison of different agent settings.

| Setting | Agent Name | Core Logic | Scope of LLM Usage | Memory | Prompting Strategy |
|---------|------------|------------|--------------------|--------|--------------------|
| 1 | Monolithic LLM | LLM | All Decisions | None | Monolithic |
| 2 | Memory-Augmented LLM | LLM | All Decisions | Last 10 Actions | Monolithic |
| 3 | Heuristic Baseline | Heuristic | None | N/A | N/A |
| 4 | Hybrid Combat Specialist | Hybrid | Combat Only | None | Monolithic (for combat) |
| 5 | Modular | Hybrid | All except | Modular | Modular |
| | Multi-Prompt Agent | | Map Routing | (Context-Specific) | |

## 3.3 Agent Architectures Under Investigation

We designed and evaluated five distinct agent architectures to systematically investigate the impact of memory, hybridization, and modularity. The specifications of each are detailed in Table 1.

- **Setting 1: Monolithic LLM Agent:** This is the baseline pure-LLM agent. It utilizes a single, comprehensive, general-purpose prompt for every decision point in the game, regardless of the context (combat, shop, event, etc.).

- **Setting 2: Memory-Augmented LLM Agent:** This architecture builds upon the Monolithic agent by incorporating a simple short-term memory buffer. The prompt includes a summary of the last 10 actions taken, each described by the situation encountered and the decision made. This directly tests the value of immediate historical context for decision-making.

- **Setting 3: Heuristic Baseline Agent:** This agent operates without an LLM. It is a fully deterministic system that follows a set of pre-programmed rules and priorities for all game situations. For example, in combat, it plays cards in a fixed priority order; at campfires, it rests if HP is below 50% and upgrades otherwise. This agent serves as a non-LLM benchmark representing a traditional, rule-based game AI.

- **Setting 4: Hybrid Combat Specialist Agent:** This is the first hybrid architecture. It uses the Heuristic Baseline agent for all non-combat decisions (map routing, shop purchases,

event choices). However, for all combat encounters, it delegates decision-making to the Monolithic LLM agent. This design tests the hypothesis that the primary value of complex LLM reasoning is concentrated within the tactically rich combat scenarios.

- **Setting 5: Modular Multi-Prompt Agent:** This is the most advanced architecture. It employs a hybrid approach, using the Heuristic Baseline for the computationally intensive task of map route planning. For all other situations, it uses an LLM, but instead of a single monolithic prompt, it deploys distinct, specialized prompts tailored to each game context (e.g., a "combat prompt," a "shop prompt," a "reward selection prompt"). This architecture tests the value of modularity and context-specificity in improving decision quality.

## 3.4 Performance Metrics

To quantitatively evaluate and compare the performance of the five architectures, we defined a primary and several secondary metrics.

- **Primary Metric: Game Progression:** The primary measure of an agent's capability is how far it can advance through the game's procedurally generated spire. This is quantified by the final Act-Level reached before the run ends (either by death or victory). Reaching a higher act or level indicates superior performance. Successfully defeating an act's boss is considered a significant milestone.

- **Secondary Metrics: remaining HP** To provide a more nuanced view of run quality, we also recorded secondary metrics. For runs that ended in failure, we logged the remaining HP of the enemy or enemies at the time of the player's defeat. For successful runs, we noted the player's final HP. These metrics help in qualitatively assessing the dominance of a victory or the narrowness of a defeat.

# 4 Empirical Results and Comparative Analysis

## 4.1 Data and Agent Mapping

The primary performance metric, game progression, revealed significant and, in some cases, counter-intuitive differences between the architectural paradigms. The results, summarized in the corrected Table 2 below, establish a clear performance hierarchy that challenges initial assumptions about the capabilities of monolithic LLM agents. Rather than a simple story of failure versus success, the data paints a more nuanced picture of competent baselines versus superior, specialized architectures.

Two findings immediately stand out. First, the baseline Monolithic LLM (Setting 1) demonstrates a surprisingly robust level of performance, achieving an average progression deep into Act 2 (2-21). This fundamentally refutes the premise that pure LLM agents are inherently brittle or tactically deficient in this complex environment. Second, and in stark contrast, the addition of a simple short-term memory buffer (Setting 2) resulted in a severe degradation of performance, making it the least successful architecture with an average progression of 1-17.

The most consistently successful agent is the Hybrid Combat Specialist (Setting 4), which not only reached the maximum possible progression (2-33) but also achieved the highest average level (2-27). This high-level result immediately suggests that the method of architectural design—specifically, the intelligent delegation of tasks—is a more critical determinant of success in Slay the Spire than the mere addition of general-purpose features like unstructured memory.

## 4.2 Overall Performance Analysis

## 4.3 The Counter-intuitive Impact of Simple Memory

A direct comparison between the Monolithic LLM (Setting 1) and the Memory-Augmented LLM (Setting 2) provides a critical insight into the value and potential pitfalls of historical context. Contrary to the common assumption that more information should lead to better decisions, the inclusion of a short-term memory buffer proved to be substantially detrimental. While the monolithic agent consistently advanced to an average level of 2-21, its memory-augmented counterpart struggled to clear Act 1, failing on average at level 1-17.

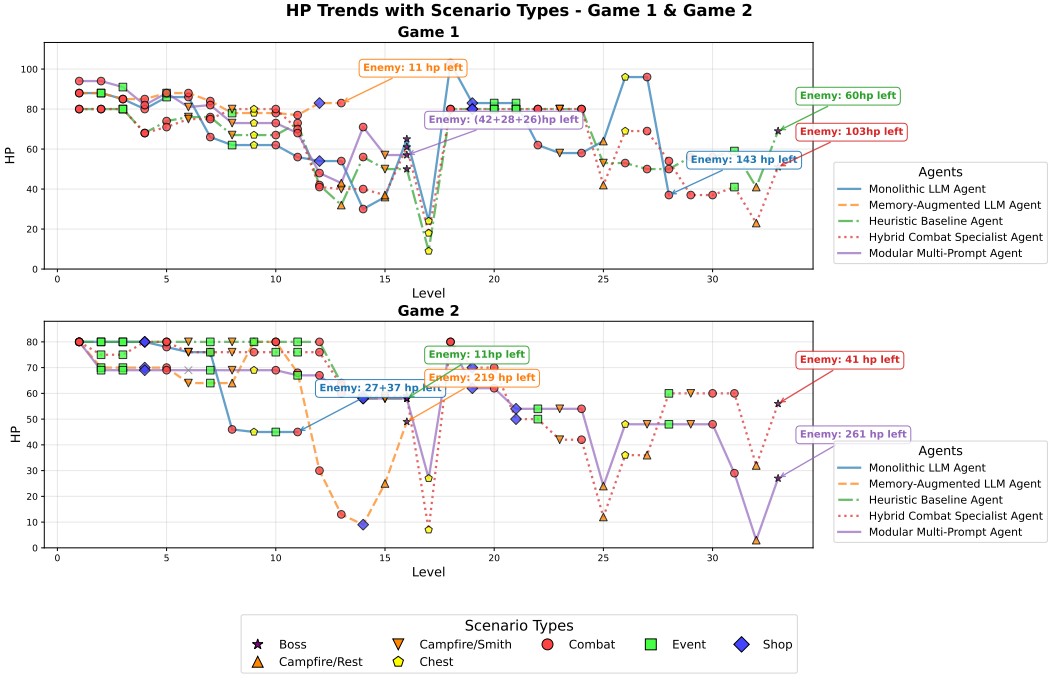

Figure 2: HP trend in game 1 and game 2

| Setting | Agent Name | Max Progression (Act-Level) | Avg. Progression (Act-Level) |
|---|---|---|---|
| 1 | Monolithic LLM | 2-28 | 2-21 |
| 2 | Memory-Augmented LLM | 2-21 | 1-17 |
| 3 | Heuristic Baseline | 2-33 | 2-22 |
| 4 | Hybrid Combat Specialist | 2-33 | 2-27 |
| 5 | Modular Multi-Prompt Agent | 2-33 | 2-22 |

Table 2: Agent progression performance across different settings.

This performance collapse indicates that the simple, unstructured provision of the last 10 actions and their contexts was not a cognitive aid but a distraction. This phenomenon can be attributed to several potential factors. The additional text may have introduced noise into the prompt, diluting the salience of the immediate game state and forcing the LLM to expend cognitive resources parsing historical data of questionable relevance. This could lead to "attentional fixation," where the agent becomes anchored to recent suboptimal plays, or a form of "contextual overload," where the model struggles to differentiate between critical real-time information and the less important historical log. This finding serves as a crucial caveat for agent design: memory is not a universally positive feature. Its implementation must be carefully structured to provide actionable insights rather than becoming a source of cognitive burden.

## 4.4 The Power of Hybridization and Heuristics

The results clearly demonstrate the superior resilience and performance ceiling of architectures that incorporate heuristic components. The three top-performing agents—the Heuristic Baseline (Setting 3), the Hybrid Combat Specialist (Setting 4), and the Modular Multi-Prompt Agent (Setting 5)—were the only architectures capable of reaching the maximum progression of 2-33, signifying their ability to complete the game's second act.

The value of combining LLM reasoning with rule-based systems is most evident in the comparison between the Heuristic Baseline and the Hybrid Combat Specialist. The heuristic agent provided a stable and respectable performance, achieving an average progression of 2-22. However, by making a single architectural change—delegating all combat decisions to the Monolithic LLM—the Hybrid

Combat Specialist's average progression increased significantly to 2-27. This five-level advancement deep within the game's most challenging act provides a clean, empirical measure of the LLM's value. It shows that in the complex, stochastic, and tactically rich domain of combat, the LLM's ability to reason about novel situations and complex synergies provides a crucial adaptive advantage that a fixed, rule-based system lacks.

## 4.5 Re-evaluating the Advantage of Modularity

The performance of the Modular Multi-Prompt agent (Setting 5) necessitates a more nuanced evaluation of modularity's benefits. A comparison between the Monolithic LLM (Setting 1, Avg: 2-21) and the Modular agent (Setting 5, Avg: 2-22) reveals nearly identical average performance. This finding contradicts the notion that modular, context-specific prompts provide a vast and universal advantage over a competent monolithic baseline. Instead, the data suggests that modularity's primary benefit is in unlocking a higher peak performance; it was one of the three architectures capable of reaching the game's final boss (Max: 2-33), a feat the monolithic agent could not achieve (Max: 2-28). The "cognitive scaffolding" provided by specialized prompts appears to be most critical not for preventing early failure, but for providing the focus needed to overcome the game's most difficult late-stage encounters.

However, an equally important comparison is between the Modular agent and the Hybrid Combat Specialist (Setting 4, Avg: 2-27). The Modular agent, which uses the LLM for more decisions (including shops and events), performed significantly worse on average. This suggests that for certain decisions, such as resource management in shops or choices in events, the simple, deterministic heuristic employed by the Hybrid Combat Specialist was more effective than the LLM's reasoning, even when guided by a specialized prompt. This implies that the optimal architecture is not one that simply modularizes all tasks for an LLM, but one that surgically delegates tasks to the component—be it an LLM or a simpler heuristic—best suited for them.

# 5 Discussion

## 5.1 Synthesizing the Findings: The Principles of Effective Strategic Delegation

This research reveals that strategic delegation is the key to designing effective LLM agents for complex domains. The success of our best-performing agent, the Hybrid Combat Specialist (Setting 4), is a direct result of this design philosophy. Rather than relying on a single component, this architecture intelligently delegates tasks. A heuristic algorithm handles the computationally heavy but strategically simple task of map navigation, which allows the LLM to be deployed where it is most valuable. This frees the LLM to apply its unique capacity for nuanced, context-dependent reasoning to the intricate and unpredictable challenges of combat. The results confirm that this division of labor—assigning the right task to the right tool—is far more effective than either a "one-size-fits-all" LLM approach or the misapplication of powerful LLM reasoning to problems that don't require it.

## 5.2 Revisiting the Tactical vs. Strategic Trade-off

Our findings provide a direct and practical revision to the challenge identified by Bateni and Whitehead, whose work suggested that LLMs possess strong strategic intuition but weak tactical execution. The robust performance of our Monolithic LLM agent (Avg: 2-21) indicates that foundational models can, in fact, exhibit considerable tactical competence without specialized architectural support. Therefore, the challenge is not necessarily to compensate for an inherent tactical weakness.

Instead, the trade-off appears to be one of cognitive load and specialization. The heuristic component in our best-performing agent acts not as a "tactical co-processor" to remedy a deficiency, but as a "specialist co-processor" that optimizes the entire system's efficiency. It handles tasks for which it is perfectly suited, allowing the LLM's "strategic core" to operate with its full reasoning capacity dedicated to the highest-value problems, such as complex combat encounters. This symbiotic relationship allows the agent to capitalize on the LLM's superior planning and adaptive abilities without being burdened by tasks that are better solved through deterministic calculation.

### 5.3 Human-Like Agents vs. Optimal Agents

Hu et al. show that LLMs approach Slay the Spire with human-like reasoning, as performance tracks human-perceived difficulty [3]. Our Hybrid Combat Specialist builds on this: not minimax-optimal, but robust—combining a steady navigation baseline with flexible, context-aware combat strategy. This human-like decision process makes it ideal for game testing and balancing, since its successes, failures, and tradeoffs mirror real players and yield more actionable feedback.

### 5.4 Limitations of the Current Study

This study, while providing clear evidence for its central claims, has several limitations that should be acknowledged. The experiments were conducted within a single, albeit complex, video game, and the findings may not generalize to all game genres. The specific heuristic baseline used was designed to be simple and representative, but a more sophisticated heuristic could alter the performance dynamics. Finally, this study observed the detrimental effect of simple memory but did not conduct a deeper analysis into the precise cognitive mechanisms causing this failure.

### 5.5 Future Directions

The architectural principles validated in this study open several promising avenues for future research.

- **Advanced Memory Systems:** The catastrophic failure of our simple memory buffer underscores the need for more sophisticated memory architectures. Future work should explore the integration of vector stores for retrieving relevant past experiences or knowledge graphs that allow agents to build a persistent, structured understanding of game mechanics and synergies, ensuring that memory serves as a strategic asset rather than a cognitive liability.

- **Investigating Failure Modes of Agentic Memory:** A direct line of inquiry should systematically investigate why and how unstructured context can harm LLM performance. This could involve controlled experiments that vary the length, structure, and content of memory buffers to identify the specific failure points, leading to a more principled approach to memory design in agentic systems.

- **Optimizing the Heuristic-LLM Boundary:** The superior performance of the Hybrid Combat Specialist over the more LLM-reliant Modular agent suggests that the boundary between tasks handled by heuristics and those handled by LLMs is a critical design choice. Future research could explore methods for automatically identifying which sub-tasks within a complex environment are best suited for each type of reasoning component, leading to the design of even more effective and efficient hybrid agents.

## 6 Conclusion

This paper presents a systematic, empirical investigation into the architectural design of LLM agents for the complex strategy game Slay the Spire, yielding a revised understanding of their capabilities. Our findings indicate that monolithic LLMs can serve as a surprisingly robust performance baseline, demonstrating significant tactical and strategic competence. However, the path to superior performance lies not in simple augmentations, such as unstructured memory, which can be counterintuitively detrimental, but in designing intelligent, modular, and hybrid frameworks built on the principle of strategic delegation.

The success of our Hybrid Combat Specialist agent, which achieved the highest average performance, provides compelling evidence for this architectural philosophy. By delegating deterministic tasks like map navigation to an efficient heuristic, the agent reserves the LLM's unparalleled semantic and strategic reasoning for the most complex and stochastic challenges of combat. This study serves as a clear demonstration that the future of agentic AI in complex domains will be defined not by the pursuit of a single, all-powerful model, but by the thoughtful integration of diverse reasoning components, paving the way for more sophisticated, resilient, and capable forms of artificial intelligence.

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

# Agents4Science AI Involvement Checklist

- **[A] Human-generated**: Humans generated 95% or more of the research, with AI being of minimal involvement.
- **[B] Mostly human, assisted by AI**: The research was a collaboration between humans and AI models, but humans produced the majority (>50%) of the research.
- **[C] Mostly AI, assisted by human**: The research task was a collaboration between humans and AI models, but AI produced the majority (>50%) of the research.
- **[D] AI-generated**: AI performed over 95% of the research. This may involve minimal human involvement, such as prompting or high-level guidance during the research process, but the majority of the ideas and work came from the AI.

1. **Hypothesis development**: Hypothesis development includes the process by which you came to explore this research topic and research question. This can involve the background research performed by either researchers or by AI. This can also involve whether the idea was proposed by researchers or by AI.

   Answer: **[C]**

   Explanation: The hypothesis begins with a human-provided idea, which the AI then expands and develops, allowing it to evolve into a complete and coherent concept.

2. **Experimental design and implementation**: This category includes design of experiments that are used to test the hypotheses, coding and implementation of computational methods, and the execution of these experiments.

   Answer: **[C]**

   Explanation: This work builds upon a publicly available community modification of the game. We extend the original repository by introducing additional player settings. The new settings—such as prompt design and game state information extraction—are implemented through AI.

3. **Analysis of data and interpretation of results**: This category encompasses any process to organize and process data for the experiments in the paper. It also includes interpretations of the results of the study.

   Answer: **[C]**

   Explanation: We manually summarized the health point (HP) trends from the game recordings and organized them into a CSV file, which was then provided to the AI. The AI evaluated the data and generated an interpretation of the results.

4. **Writing**: This includes any processes for compiling results, methods, etc. into the final paper form. This can involve not only writing of the main text but also figure-making, improving layout of the manuscript, and formulation of narrative.

   Answer: **[D]**

   Explanation: The paper was generated by AI. We provided the AI with our data, relevant literature, and prompts instructing it to follow the rubric established in the Agent4Science AI paper. We subsequently reviewed the draft and made only minimal modifications.

5. **Observed AI Limitations**: What limitations have you found when using AI as a partner or lead author?

   Description: During planning, the agents often produce errors due to excessively long contextual information, which requires human intervention to correct or provide guidance.

